# Microscopic Kinetics in Poly(Methyl Methacrylate) Exposed to a Single Ultra-Short XUV/X-ray Laser Pulse

**DOI:** 10.3390/molecules26216701

**Published:** 2021-11-05

**Authors:** Nikita Medvedev, Jaromír Chalupský, Libor Juha

**Affiliations:** 1Department of Radiation and Chemical Physics, Institute of Physics, Czech Academy of Sciences, Na Slovance 2, 182 21 Prague 8, Czech Republic; chal@fzu.cz (J.C.); juha@fzu.cz (L.J.); 2Laser Plasma Department, Institute of Plasma Physics, Czech Academy of Sciences, Za Slovankou 3, 182 00 Prague 8, Czech Republic

**Keywords:** PMMA, free-electron laser, ablation, nonthermal melting, band gap collapse

## Abstract

We study the behavior of poly(methyl methacrylate) (PMMA) exposed to femtosecond pulses of extreme ultraviolet and X-ray laser radiation in the single-shot damage regime. The employed microscopic simulation traces induced electron cascades, the thermal energy exchange of electrons with atoms, nonthermal modification of the interatomic potential, and a triggered atomic response. We identify that the nonthermal hydrogen decoupling triggers ultrafast fragmentation of PMMA strains at the absorbed threshold dose of ~0.07 eV/atom. At higher doses, more hydrogen atoms detach from their parental molecules, which, at the dose of ~0.5 eV/atom, leads to a complete separation of hydrogens from carbon and oxygen atoms and fragmentation of MMA molecules. At the dose of ~0.7 eV/atom, the band gap completely collapses indicating that a metallic liquid is formed with complete atomic disorder. An estimated single-shot ablation threshold and a crater depth as functions of fluence agree well with the experimental data collected.

## 1. Introduction

Poly(methyl methacrylate) (PMMA) is a polymer that is widely used in the free-electron laser experiments for pulse characterization via ablation [1] and desorption imprints [2]. The material hardness and robustness to mechanical stress together with sensitivity to radiation damage makes it a perfect candidate for applications, such as pulse monitors. Despite its wide experimental use during the last two decades, it still remains very challenging to study it theoretically [2,3,4]. A lack of understanding of the microscopic details of damage hinders further development of the techniques employing PMMA.

In models employed in the community, the processes playing a role in the damage often need to be chosen ad hoc [4,5]. For example, often an assumption of thermal melting is used to evaluate the damage threshold [6], or nonthermal bond breaking is assumed to be the dominant damage mechanism [7]. A choice of proper models requires microscopic understanding of processes involved, which is still lacking in the field of complex materials, such as polymers [3]. It is especially true with respect to irradiation with modern free-electron lasers, such as FLASH [8], FERMI [9], LCLS [10], SACLA [11], and European XFEL [12]. They provide femtosecond high-intensity extreme ultraviolet (XUV) and X-ray pulses capable of single-shot material ablation, the nature of which remains an open question.

Here, we apply a detailed model to simulate the PMMA response to ultrafast XUV and X-ray exposure in the regime of single-shot damage [13,14]. We study the electronic and atomic behavior of irradiated PMMA targets to identify the nature of the damage. The model includes the following stages: (a) nonequilibrium electron cascades, (b) coupling of electrons and atoms (electron–phonon coupling) possibly leading to thermal melting, (c) modification of the interatomic potential energy surface due to electronic excitations, and (d) atomic relocation triggered by such a change in the interatomic forces, which may result in nonthermal bond breaking and melting. Tracing all the involved processes allows us to identify their relative importance and reveal the microscopic kinetics of damage formation in PMMA at various deposited doses.

## 2. Model

To model femtosecond irradiation of PMMA, we used XTANT-3 hybrid code describing electronic kinetics and atomic dynamics with changing interatomic potential and coupling between electrons and ions [13]. XTANT-3 includes a few interconnected models that are executed in parallel and exchange information on each time step of the simulation:A Monte Carlo (MC) simulation is used to model the XUV or X-ray photon absorption and induced electron cascades, including all secondary electrons and Auger decays of core holes [15]. An excited electron is traced until its energy falls below a predefined cut off of 10 eV counted from the bottom of the conduction band of the material, the lowest unoccupied molecular orbital (LUMO). Elastic electron scattering, transferring kinetic energy to atoms, is described with Mott’s cross section with a modified Molier screening parameter [16]. Inelastic electron scattering (impact ionization) is described with binary-encounter Bethe (BEB) scattering cross section [17].Low-energy electrons populating the valence band and the conduction band below the cut off energy are assumed to follow Fermi–Dirac distribution at all times [13]. This fraction of electrons loses electrons due to photo- and impact ionizations and gains new particles when an electron from the MC model loses its energy below the cut off [15]. The energy of such an electron is then added to the low-energy fraction of electrons, leading to a change in its chemical potential and temperature. This fraction of electrons also exchanges energy with the atoms via the nonadiabatic coupling described below.Nonadiabatic coupling between electrons and ions is calculated with the Boltzmann collision integral [18]. This allows us to model the evolution of the system beyond the Born-Oppenheimer approximation. In case of periodic atomic motion within an ideal crystal, it reduces to the electron–phonon coupling. However, the applied model is capable of describing coupling of electrons to an arbitrary atomic displacement beyond the phononic approximation [19]. The matrix elements entering the Boltzmann integrals are obtained with the use of the tight binding molecular dynamics simulations similar to the Tully surface hopping method [20]. Transient electron populations are defined by the above-mentioned Fermi–Dirac distribution.The transferable tight binding (TB) method is used to describe the evolution of the molecular orbitals (electronic density of states), interatomic forces, and matrix elements for nonadiabatic electron–ion coupling. In the present work, we employ the density-functional tight binding (DFTB) method with matsci-0-3 parameterization on the non-self-consistent level [21]. This parameterization uses an sp^3^ linear combination of atomic orbitals (LCAO) basis set for hydrogen, carbon, and oxygen atoms [22]. The TB Hamiltonian depends on the positions of all the atoms in the simulation box and, thus, evolves on each time step of the simulation. The TB parameterization does not depend on the electronic temperature, which is an approximation; however, it has been demonstrated in a series of previous works that such a model describes irradiated systems with a reasonable accuracy (see overview [13] and the references therein, showing comparisons with available FEL experiments).Atomic motion is traced with help of the classical Molecular Dynamics (MD) method. It solves Newtonian equations of motion for each atom using the Verlet algorithm [23] with the potential energy surface provided from the TB method described above. Additionally, energy gain or loss from the nonadiabatic coupling of low-energy electrons and elastic scattering of high-energy electrons is fed to atoms via velocity scaling at each time step. We used 0.1 fs time steps in the simulations. A microcanonical (NVE) ensemble was used with periodic boundary conditions.

All further details of the code can be found in [13]. To model PMMA, we used 240 atoms in the supercell. Prior to productive simulation runs, the supercell was relaxed using a zero-temperature molecular dynamics (the steepest descent method) simulation to find the equilibrium atomic positions and supercell vectors for the given interatomic potential. Figure 1 shows this prepared initial atomic configuration, which mimics suspended strands of PMMA molecules, similarly to previous works on the modeling of polymers under periodic boundary conditions [24,25]. Despite the fact that PMMA may contain a wide variety of strands arrangements or even be in amorphous phase [26], here, we chose the simplest and most accessible configuration for tight-binding modelling. 

The simulation starts from this initial configuration with random velocities assigned to atoms according to the Maxwellian distribution at the room temperature. Atoms are allowed to thermalize for a few hundred femtoseconds prior to arrival of the laser pulse centered at *t* = 0 fs. The laser pulse used is Gaussian with 10 fs FWHM and 92 eV photon energy. After irradiation, the system is traced up to 2 picoseconds to observe the damage kinetics at sufficiently high deposited doses. Many simulation runs are performed with the absorbed doses ranging from 0.01 to 1 eV/atom to identify various damage thresholds in the system. Atomic snapshot illustrations are prepared with help of VMD software [27].

## 3. Results

The simulation results show that PMMA started to exhibit the first signs of damage in the bulk at the energy density around 0.07 eV/atom. Figure 2 clearly demonstrates that first stable defect levels formed only at doses above ~0.07 eV/atom, indicating that no defects were formed at lower doses. At the shown dose of 0.06 eV/atom, only a transient perturbation of the energy levels occurred, quickly recovering and forming no lasting damage. At higher doses, the defect levels within the gap persisted throughout the simulation time. This allowed us to unambiguously identify the damage threshold. This takes place via the following processes. 

Some hydrogen atoms begin to detach from parental carbons and may attach to the neighboring strain. Concurrently, active processes of cross-linking start. Hydrogen detachment also leads to local bond breaking, resulting in the fragmentation of MMA molecules, as was also discussed, e.g., in [5]. An example of these processes is shown in Figure 3 for the deposited dose of ~0.1 eV/atom. We see that the final state consists of a few MMA molecules broken into molecular fragments, which then diffuse between neighboring PMMA strains.

The maximal electronic temperature under these conditions reaches ~10,000 K, see Figure 4. The electron–ion (electron–phonon) coupling rises up to ~6 × 10^17^ W/(m^3^K) at the peak of the electronic temperature, but quickly drops to below 1 × 10^17^ W/(m^3^K) (Figure 4). The electron–ion energy exchange leads to an atomic temperature increase up to about 800 K by the time of 2 ps. Full equilibration of the electronic and the atomic temperatures does not occur within this time. Note that some oscillations in the electron temperature are present due to changes in the material band gap that affect electronic populations. Spikes in the coupling parameter, albeit strong, average out at different time steps and have only a minor effect on the temperatures as can be seen in Figure 4.

During the damage process, there is a transient band gap collapse as seen in Figure 4, accompanied by detachment of hydrogen atoms and fragmentation of some MMA molecules. This indicates that the transition takes place via the electronically conducting phase, ending up in a semiconducting phase with a small band gap present. To better understand the changes in electronic structure, we plot the evolution of the electronic energy levels (eigenstates, or molecular orbitals) in Figure 5. This figure demonstrates that the highest occupied molecular orbital (HOMO) and LUMO states meet shortly after the FEL pulse, and a few more levels join together soon after. 

This creates defect levels within the gap, while the majority of the energy states are still within the well-separated valence and conduction bands. However, the presence of electrons on these created defect levels (formed by the raised HOMO levels) and availability of the unoccupied states (formed by lowered LUMO levels) make the irradiated PMMA transiently semiconducting. This happens as the lowest level of the conduction band (LUMO) transiently joins the defect levels within the band gap produced by raising the HOMO level from the valence band. 

They then experience an avoided crossing at the times of a few tens of fs. Two more levels join in soon, at the times of ~100 to ~150 fs. However, at the time of ~200 fs, one level returns to the bottom of the conduction band. All this affects the width of the band gap, since it is defined as the difference between the HOMO and LUMO levels, which are highly dynamic at these times. The ongoing complex atomic dynamics (Figure 3), showing the connection and disconnection of different PMMA strands, is accompanied with corresponding changes in the band structure.

In this process of damage, atomic charges are practically unaffected as Mulliken analysis shows [28] with only a slight redistribution of electrons from carbon to oxygen atoms (Figure 4). Mulliken charges are defined by the electronic populations on different kinds of energy levels (molecular orbitals of different kinds of atoms). Even though the energy levels are shifting during the phase transition, the electronic occupations are not drastically affected (electrons are redistributed among the levels of the same kind), leading to only minor changes in the average charges.

Hydrogen atoms have practically the same charge of around +0.16 during the entire simulation. This suggests that, in a near surface region, where hydrogen atoms may be emitted under irradiation, the majority of them should be observed as neutral atoms, with a presence of about 16% charged ions (protons). So, an emission of molecular fragments may be expected, which should lead to surface ablation as will be discussed in the next section.

With an increase of the dose (fluence), more hydrogen atoms detach and reattach to other carbons and oxygens, increasing the disorder. At the dose of ~0.5 eV/atom, the MMA molecules start to actively break into atomic species; see Figure 6. The carbon atoms form chains, whereas some oxygen atoms are still attached to their parenting carbons or float freely. During this damage, the band gap of PMMA shrinks but does not fully collapse (see Figure 7), indicating that it is still in a semiconducting state.

The behavior of the electronic structure after 0.5 eV/atom dose deposition is shown in Figure 8. One can see that, in addition to the formation of the defect levels within the gap (similarly to the abovementioned case of 0.1 eV/atom, cf. Figure 5), the bands widened, shrinking the gap. It is rather difficult to distinguish the “defect” levels from the band levels at such doses. With further increase of the dose, the bands completely merge: a full band gap collapse occurs at higher doses of ~0.7 eV/atom, turning PMMA into a fully disordered metallic liquid.

After 0.5 eV/atom dose deposition, the temperatures of different elements do not fully coincide but rise with different rates (see Figure 7). Due to nonthermal effects, electronic and atomic temperatures equilibrate fast, significantly faster than we observed above for 0.1 eV/atom dose, while the electron–phonon coupling parameter is only about three-times higher at its peak (cf. Figure 7 vs. Figure 4; the same effect was observed in polyethylene in [25]). 

The temperature of oxygen atoms here rises slightly faster than that of carbon and hydrogen atoms. This appears to indicate the kinetic pathway of damage formation: the modification of interatomic forces seems to affect oxygen bonds the most. This additional force acting on oxygen atoms increases their kinetic energy faster than that of other atoms. However, oxygen bonds are stronger than hydrogen bonds, so while hydrogen detaches from parenting atoms, oxygen may still be attached to carbons.

During the damage, there is again only a slight redistribution of electrons between the parental atoms: as Figure 7 shows, during the excitation stage, a small fraction of electrons transfers from carbon to oxygen atoms, which then quickly returns during the fast electronic temperature decrease. After around ~1.5 ps, there is a secondary electron redistribution due to ongoing damage and separation of different elements indicating a presence of active chemical reactions. We can see carbon atoms forming mainly linear chains, while oxygen atoms and hydrogen atoms are grouping among themselves (Figure 6). The hydrogen average charge remains almost unaffected, increasing from 0.16 to only ~0.18 at the end of the simulation for a 0.5 eV/atom deposited dose. This suggests that, at such irradiation parameters, the charge states of emitted hydrogens should not be very sensitive to the FEL fluence.

## 4. Discussion

Although the simulations were performed in the bulk applying periodic boundary conditions to the supercell, the damage predicted to take place in PMMA allows interpretation of it as a cause of ablation of the surface. Indeed, the detachment of hydrogen atoms and molecular fragments at the deposited dose above ~0.07 eV/atom indicates that those free-floating fragments and atoms should be able to leave the surface. We, therefore, convert our threshold dose into an estimate of an incoming threshold fluence, according to the relation [13]: $F_{th}=D_{th}n_{at}l_{at}$, where *F_th_* is the threshold fluence, *D_th_* is the evaluated threshold dose, *n_at_* is the atomic density of PMMA, and $l_{at}$ is the photon attenuation length for the given photon energy (taken from [29]). 

This relation assumes normal incidence of the FEL pulse and that the damage is defined solely by the deposited energy density and not sensitive to a particular photon energy. Since core holes in the K-shell of carbon and oxygen atoms decay within a few femtoseconds [30], their transient presence should not change the damage mechanism and, thus, would quickly result in conditions similar to those induced by photon energies below K-edges [31]. This allows us to convert the calculated absorbed dose into the incoming fluence across a wide range of photon energies.

The resulting threshold fluence is compared to the available experimental data in Figure 9. We see in this figure that the agreement between the estimated threshold fluence and the experimental one is reasonable. Hosaka et al.’s data point on the ablation threshold of PMMA [32] is an order of magnitude lower than the predicted curve and than the other experimental data at the same photon energy (92 eV). This discrepancy is attributed to the fact that Hosaka et al. used a different experimental photon source with 7 ps pulse duration rather than ultrashort FEL pulses used in all other experiments collected in Figure 9.

A reasonable agreement between the theoretically calculated damage threshold fluence and the experimentally measured ablation threshold supports our interpretation of the results. We, thus, conclude that the onset of the damage in the bulk of PMMA is accompanied by the start of the single-shot ablation at the surface.

Considering the same nature of the damage in the bulk and at the surface of PMMA, we may evaluate the maximum depth of the ablation crater. To do that, we assume that the crater depth profile is determined by the iso-surface corresponding to the depth where the damage threshold dose is deposited, as was also suggested in [2]. Then, as it follows from the Lambert–Beer law of radiation absorption, the maximum crater depth *d_max_* may be estimated as $d_{max}=l_{at}\;\ln\left(D/D_{th}\right)$, where *D* is the absorbed dose corresponding to the peak fluence of an incoming FEL pulse. 

The resulting dependence of the ablation depth on the deposited dose for the photon energy of 92 eV is shown in Figure 10. For the estimate here, we used the theoretical value of the damage threshold dose found above, *D_th_* = 0.07 eV/atom (which corresponds to ~1.3 kJ/cm^3^ in PMMA), and thus this estimate is fully predictive. We see that the theoretical curve coincides very well with the experimental data from [2]. This validates our findings and conclusions on the damage in PMMA and supports the interpretation of the nature of the single-shot damage reported in [2,5].

## 5. Conclusions

The response of PMMA to irradiation with ultrafast FEL pulses was studied theoretically with help of the hybrid code XTANT-3. Accounting for thermal electron–ion coupling, tracing atomic dynamics, including nonthermal evolution of the electronic structure and induced bond breaking, allowed us to analyze various damage channels and their interplay. We demonstrated that the damage threshold in PMMA is ~0.07 eV/atom (1.3 kJ/cm^3^) at which the first detachment of hydrogen atoms takes place leading to PMMA strains cross-linking and molecular fragmentation.

This damage is a result of nonthermal effects caused by electronic excitation. It proceeds with a transient collapse of the band gap at femtosecond timescales via formation of defect levels, subsequently reopening to a value of ~1 eV, which indicates a semiconducting damaged state. The electron–ion coupling at such doses is too slow and heats up the atomic system only at later timescales of a few picoseconds.

With an increase of the deposited dose above ~0.5 eV/atom, the fragmentation into atoms and atomic disorder takes place due to nonthermal melting. At doses above ~0.7 eV/atom, the disordered state turns metallic with a complete band gap collapse. In all cases, the damage has a nonthermal nature, whereas the thermal contribution is only minor.

The estimated damage threshold was compared with the experimental ablation threshold, showing a reasonable agreement. This result supported our conclusion that the damage threshold in the bulk coincides with the onset of ablation at the surface, as the detachment of atoms and molecular fragmentation is expected to yield material emissions. An evaluated maximum ablation crater depth as a function of a deposited dose for 92 eV photon energy irradiation showed excellent agreement with the experimental data, further validating our model and the conclusions drawn.

## Figures and Tables

**Figure 1 molecules-26-06701-f001:**
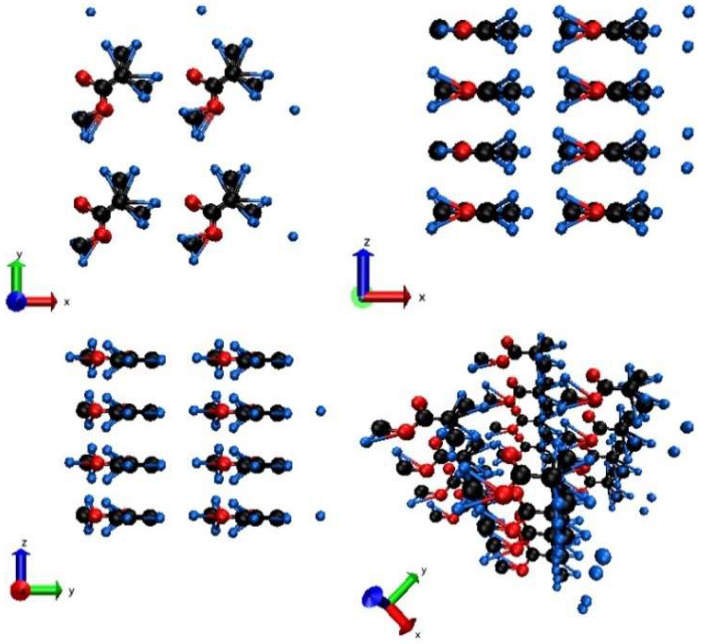
Initial atomic structure of PMMA in the supercell: top view along Z (**top left**), side view along Y (**top right**), side view along X (**bottom left**), perspective view (**bottom right**). Black balls are C, red balls are O, and blue balls are H atoms.

**Figure 2 molecules-26-06701-f002:**
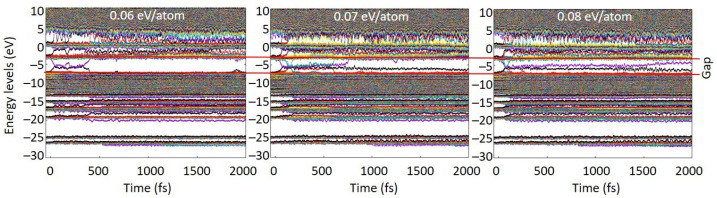
Evolution of the electronic energy levels (eigenstates; molecular orbitals) in PMMA irradiated with 0.06 eV/atom (left panel), 0.07 eV/atom (middle panel), and 0.08 eV/atom (right panel) deposited doses. Initial HOMO-LUMO gap is marked with red lines to illustrate formation of defect levels within the gap.

**Figure 3 molecules-26-06701-f003:**
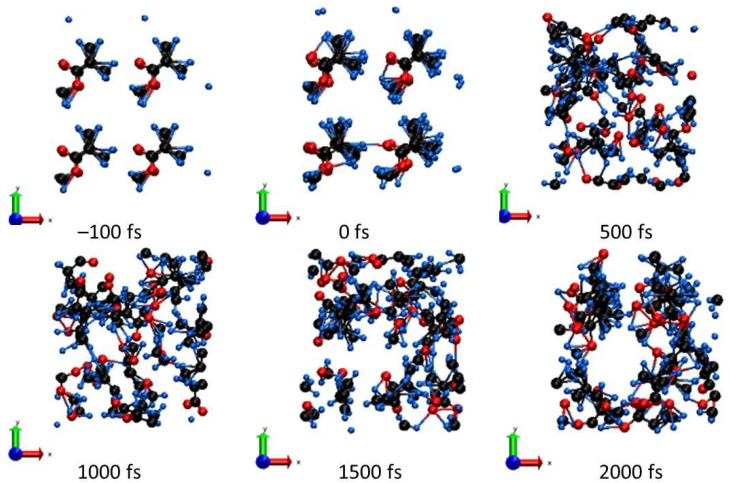
Atomic snapshots of the PMMA supercell irradiated with 0.1 eV/atom at different time instants.

**Figure 4 molecules-26-06701-f004:**
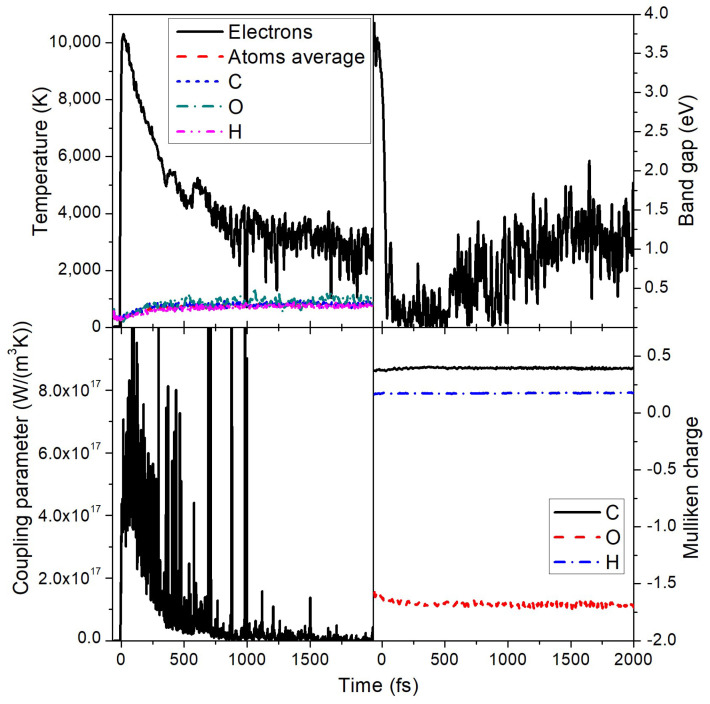
Several quantities in PMMA irradiated with a 0.1 eV/atom deposited dose. (**Top left panel**) Electronic and atomic temperatures, together with partial temperatures of carbon, oxygen, and hydrogen atomic subsystems. (**Top right panel**) Band gap of PMMA. (**Bottom left panel**) Electron–ion (electron–phonon) coupling parameter evolution during and after irradiation. (**Bottom right panel**) Mulliken charges on carbon, oxygen, and hydrogen ions.

**Figure 5 molecules-26-06701-f005:**
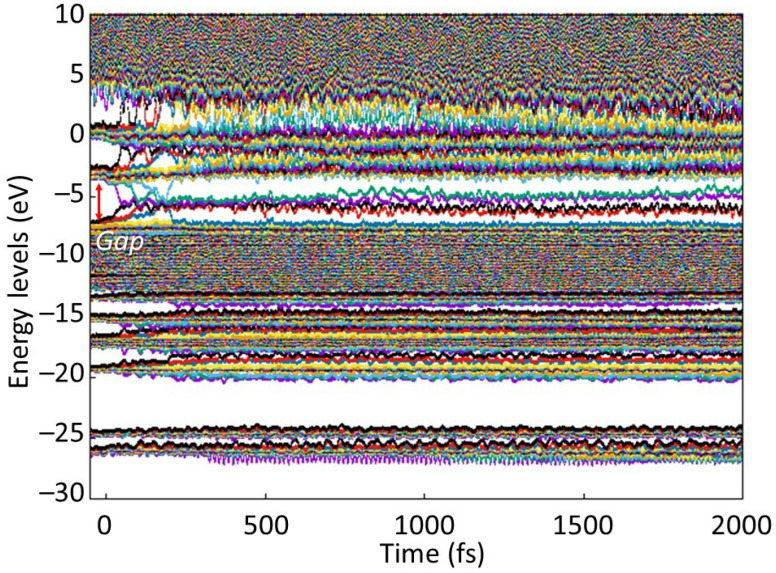
Evolution of the electronic energy levels (eigenstates; molecular orbitals) in PMMA irradiated with 0.1 eV/atom deposited dose. The initial HOMO-LUMO gap is marked with a red arrow.

**Figure 6 molecules-26-06701-f006:**
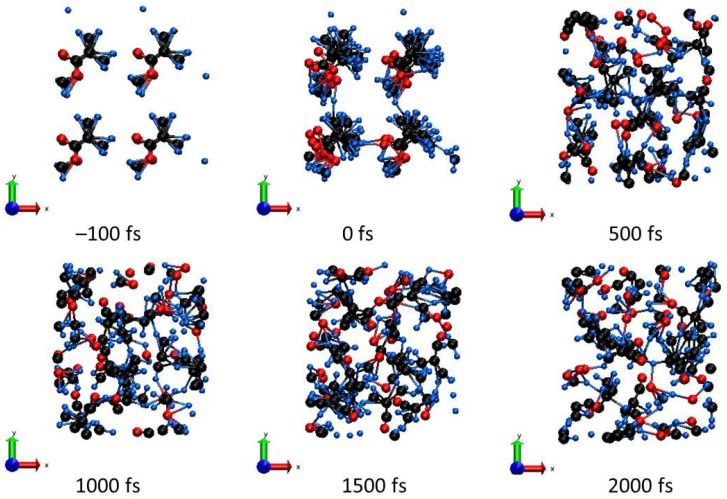
Atomic snapshots of the PMMA supercell irradiated with 0.5 eV/atom at different time instants.

**Figure 7 molecules-26-06701-f007:**
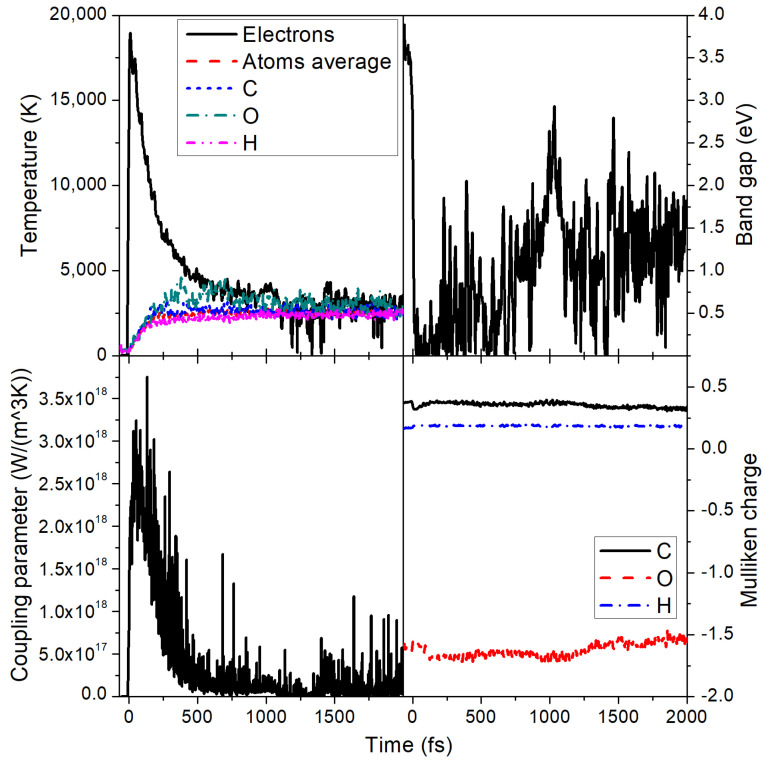
Several quantities in PMMA irradiated with 0.5 eV/atom deposited dose. (**Top left panel**) Electronic and atomic temperatures, together with partial temperatures of carbon, oxygen, and hydrogen atomic subsystems. (**Top right panel**) Band gap of PMMA. (**Bottom left panel**) Electron–ion (electron–phonon) coupling parameter evolution during and after irradiation. (**Bottom right panel**) Mulliken charges on carbon, oxygen, and hydrogen ions.

**Figure 8 molecules-26-06701-f008:**
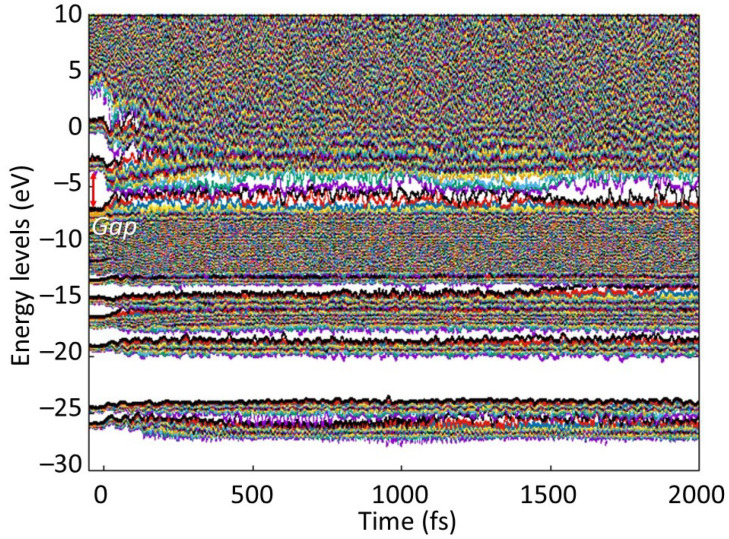
Electronic energy levels (eigenstates) in PMMA irradiated with 0.5 eV/atom deposited dose. HOMO-LUMO gap is marked with a red arrow.

**Figure 9 molecules-26-06701-f009:**
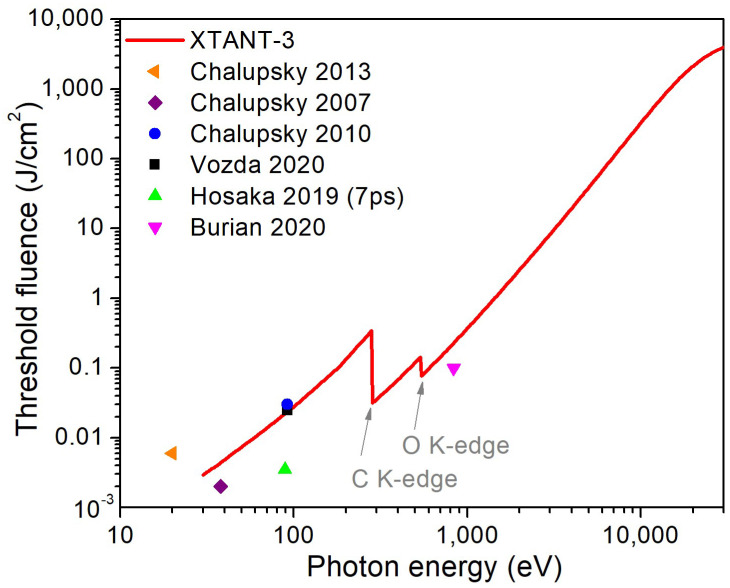
The damage threshold fluence in PMMA as a function of incoming photon energy estimated with XTANT-3 and available experimental data [1,2,5,32,33,34].

**Figure 10 molecules-26-06701-f010:**
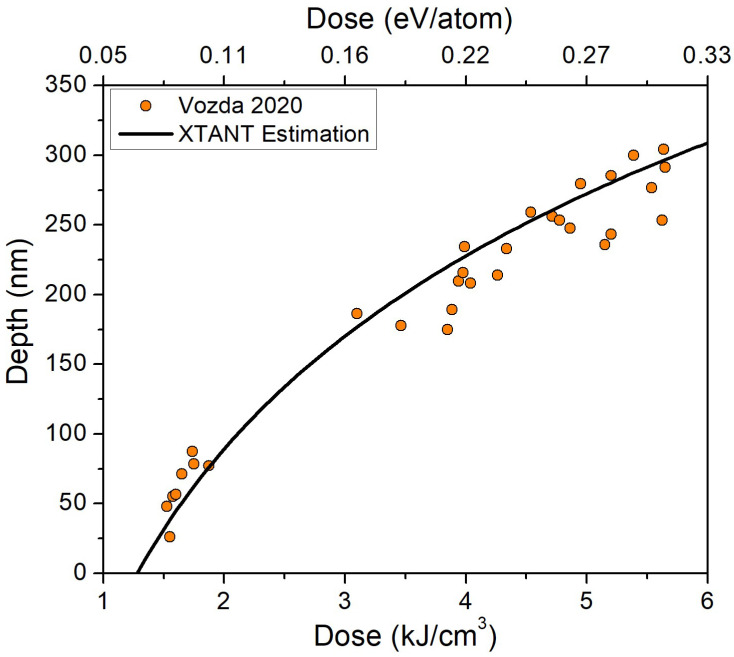
Estimated maximum single-shot ablation crater depth in PMMA as a function of the peak absorbed dose at 92 eV photon energy, as estimated with XTANT-3 and compared to experimental data from [2].

## Data Availability

Data produced in this work may be available from the corresponding author upon reasonable request.

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
