# Peer review of "Microscopic Kinetics in Poly(Methyl Methacrylate) Exposed to a Single Ultra-Short XUV/X-ray Laser Pulse"

_molecules, 2021, doi:10.3390/molecules26216701_

Round 1
Reviewer 1 Report
The authors theoretically studied the response of PMMA to irradiation with ultrafast FEL pulses by the hybrid code XTANT-3. They found the damage threshold in PMMA is ~0.07 eV/atom, the band gap transiently collapse at femtosecond timescales via formation of defect levels… I can recommend this manuscript for the publication in Molecules as an article, but I still have some questions.
1、 This manuscript found the damage threshold of PMMA is 0.07 eV/atom, I think showing some figure to prove this threshold would be better. I suggest to illustrate the dependence of the ratio of detaching hydrogen on the input absorbed does.
2、 It is hard to understand the description: “modification of interatomic potential energy surface due to electronic excitations and atomic relocation triggered by such a change in the interatomic forces possibly resulting in nonthermal bond breaking and melting. ”
3、 The statement:” In this process of damage, atomic charges are practically unaffected as Mulliken analysis shows [28], with only a slight redistribution of electrons from carbon to oxygen atoms 159 (Figure 3)”, the PMMA has undergone a transient band gap collapse, why the atomic charges are still practically unaffected?
4、 The pulse width of the driving laser is 10 fs, Fig.3 infers the PMMA become a transient semiconductor at ~200fs, which mechanism may invoked this delay?
Author Response
Please see the file attached

Reviewer 2 Report
The manuscript describes numerical simulations of microscopic ablation of PMMA by free electron laser soft x-ray radiation in sub-fs to ps time scale using metods of molecular dynamics. Various nonthermal processes of PMMA damage were analyzed in the range of absorbed radiation doses 0.07–0.7 eV/atom. The lowest dose causing PMMA disruption was found and compared with the available experimental data.
As a comment there are some abbreviations in the text which might be not known to the inexperienced reader, e.g. LUMO (row 58), DFTB (row 82), LCAO (row 84), HOMO (row 142). It would be better to reveal the meaning of abbreviations as it has been already done with the others.
Author Response
We expanded all the abbreviations in the revised text.